

# Update of machine learning for ultrasound diagnosis of metabolic dysfunction-associated steatotic liver disease: a bright future for deep learning

Jiawen Li[1,*], Jianhui Chen[1,*], Xiaohong Zeng[2], Guorong Lyu[1], Shu Lin[3,4] and Shaozheng He[1]

[1] Department of Ultrasound, the Second Affiliated Hospital of Fujian Medical University, Quanzhou, Fujian Province, China
[2] Department of Health Care, the Second Affiliated Hospital of Fujian Medical University, Quanzhou, Fujian Province, China
[3] Centre of Neurological and Metabolic Research, the Second Affiliated Hospital of Fujian Medical University, Quanzhou, Fujian Province, China
[4] Group of Neuroendocrinology, Garvan Institute of Medical Research, Sydney, New South Wales, Australia
[*] These authors contributed equally to this work.

Corresponding authors
Shu Lin, shulin1956@126.com
Shaozheng He, 1251282489@qq.com

## ABSTRACT

Metabolic dysfunction-associated steatotic liver disease (MASLD) is the most common liver disease and the burden is increasing around the world. Ultrasound diagnosis of MASLD is the preferred method due to its convenience, absence of radiation, and high accuracy. The application of artificial intelligence (AI) in MASLD diagnosis has been explored across electronic medical records, laboratory tests, ultrasound and radiographic imaging, and liver histopathological data. Notably, AI's application in ultrasound diagnosis has garnered significant attention. Deep learning (DL), known for its exceptional image recognition and classification capabilities, has become a focal point in ultrasound research. This paper reviews and analyzes the application of various machine learning (ML) algorithms in the ultrasound diagnosis of MASLD, highlighting the advantages and potential of AI in this field. It is intended for clinicians, AI researchers, and healthcare innovators, aiming to enhance diagnostic accuracy, expand MASLD screening in primary care, and support early diagnosis, prevention, and treatment.

## INTRODUCTION

Non-alcoholic fatty liver disease (NAFLD) is the most prevalent liver disease worldwide, with a global prevalence of 38% (*Wong et al., 2023*). NAFLD is characterized by the accumulation of fat in more than 5% of hepatocytes, in association with metabolic risk factors, particularly obesity and type 2 diabetes, and the absence of excessive alcohol consumption ($\geq$30 g/day in men and $\geq$20 g/day in women) or other chronic liver diseases.

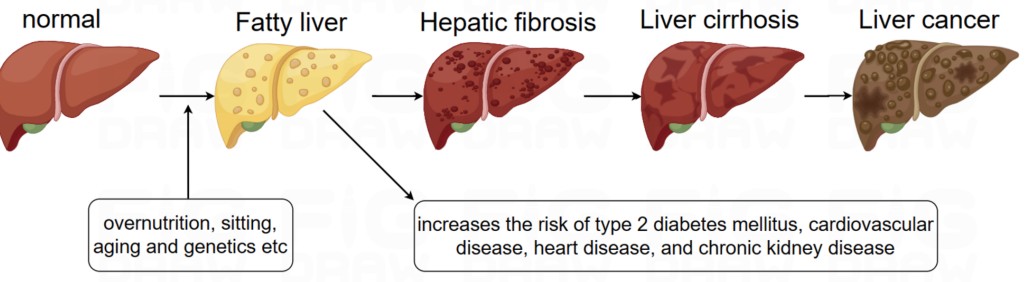

**Figure 1 Evolution of NAFLD.**

The condition encompasses a spectrum of liver disorders, ranging from simple steatosis, which may or may not involve mild inflammation, to non-alcoholic steatohepatitis (NASH) (*Powell, Wong & Rinella, 2021*). It is increasingly recognized that NAFLD is a multisystem disease, where insulin resistance and associated metabolic dysfunction play a key role in its development and in the progression of liver-related morbidities, including cirrhosis, liver failure, and hepatocellular carcinoma (HCC), as well as extrahepatic complications such as cardiovascular disease (CVD), type 2 diabetes mellitus (T2DM), chronic kidney disease (CKD), and certain types of extrahepatic cancers (*Byrne & Targher, 2015*; *Devarbhavi et al., 2023*). Although the progression of NAFLD can lead to cirrhosis, liver failure, and HCC, most deaths in NAFLD patients are due to CVD (Fig. 1). In 2023, three major multinational liver associations recommended replacing the term NAFLD with metabolic dysfunction-associated steatotic liver disease (MASLD), and the term metabolic dysfunction-associated steatohepatitis (MASH) was proposed to replace NASH. The diagnostic criteria for MASLD involve the presence of hepatic steatosis (identified through imaging or biopsy) alongside at least one cardiometabolic risk factor (CMRF), with no other identifiable causes of liver steatosis (*Rinella et al., 2023a*). Emerging evidence suggests a near-perfect concordance between the definitions of NAFLD and MASLD—approximately 99% of individuals with NAFLD meet MASLD criteria (*Targher, Byrne & Tilg, 2024*).

Auxiliary screening methods for MASLD mainly include ultrasonography, magnetic resonance imaging (MRI), computed tomography (CT), laboratory tests, and pathological tests. Ultrasound technology is the first choice for imaging examination of MASLD because it is real-time, inexpensive, non-invasive, and non-radioactive. Accurate ultrasound diagnosis of MASLD requires an experienced physician, as it has a subjective component and can be affected by instrument noise and speckle. Ultrasound-based AI has already been used for various applications, such as thyroid, breast, and liver-related diseases. Currently, AI for ultrasound in MASLD is a research hotspot, which has evolved from quantitative ultrasound (QUS) to machine learning (ML) to deep learning (DL), with increasing diagnostic accuracy. In this review, we will discuss the application of ultrasound-based machine learning in the diagnosis of MASLD, focusing on the research and prospects of deep learning, with a view to its wide clinical application.

## SURVEY METHODOLOGY

To conduct an unbiased review of machine learning applications in ultrasound diagnosis of metabolic dysfunction-associated steatotic liver disease (MASLD), the following approach was used:

**Search engines and databases:** PubMed, Web of Science, Scopus.

**Search terms:** ("machine learning" OR "deep learning") AND ("ultrasound" OR "ultrasonography") AND ("NAFLD" OR "MASLD"), including relevant synonyms, related terms, and MeSH terms.

**Inclusion criteria:** Peer-reviewed original English articles from 2010 to 2024, focusing on machine learning in ultrasound for MASLD diagnosis or evaluation, with quantitative diagnostic data (*e.g.*, accuracy, sensitivity, specificity).

**Exclusion criteria:** Review articles, editorials, non-original research, studies using non-ultrasound imaging modalities (*e.g.*, MRI, CT), or not validating machine learning models for MASLD.

**Potential bias:** Reliance on indexed databases may exclude non-English or non-indexed studies. Publication and reviewer bias may affect the selection of studies.

**Screening process**: Two independent reviewers screened titles and abstracts, and full-text articles were assessed based on the criteria. Disagreements were resolved through discussion, with a third reviewer consulted if needed.

**Contributions:** This review differs from existing literature by providing a comprehensive comparison of various machine learning models applied in the ultrasound diagnosis of metabolic MASLD, with a particular focus on the advancements in deep learning technologies. This review covers a wide range of models, from classical machine learning techniques to advanced deep learning architectures, highlighting their respective strengths and challenges. Additionally, the review offers insights into the future clinical applications of deep learning in MASLD diagnosis and outlines key areas for future research.

### Metabolic dysfunction-associated steatotic liver disease

Although MASLD can be diagnosed by imaging tests such as ultrasound, CT, or MRI, diagnosing MASH still requires a liver biopsy to identify features such as inflammation, hepatocyte ballooning, Mallory-Denk bodies, and early fibrosis (*Neuschwander-Tetri, 2017*). Furthermore, early identification, prevention, and treatment of MASLD can significantly reduce its serious consequences.

Conventional ultrasound is the first choice for diagnosing fatty liver disease. This non-invasive technique is frequently used to screen for MASLD and is recommended by the European Guidelines for MASLD Management as the first-line imaging method for patients at risk (*Miele et al., 2020*). The typical ultrasound features of MASLD include liver echogenicity higher than the right kidney, with varying degrees of distal attenuation and intrahepatic vascular blurring. Based on these characteristics, the degree of steatosis can be subjectively classified as mild, moderate, or severe. Traditional B-mode ultrasonography has high diagnostic accuracy for moderate and severe hepatic steatosis ($\geq$20% hepatic fat) but low diagnostic accuracy for mild steatosis (<20% hepatic fat) (*Dasarathy et al., 2009*; *Lee et al., 2010*) and does not reliably detect individuals with a high body mass index (BMI) (>40

kg/m$^2$). Ultrasound is inexpensive, versatile, non-invasive, and non-radioactive. Current European practice guidelines recommend identifying hepatic steatosis by ultrasound, as it is more widespread and cheaper than the gold standard and MRI (*European Association for the Study of the Liver (EASL), 2016*).

## Quantitative ultrasound in diagnosis of MASLD

Advances in ultrasound technology have significantly enhanced the diagnosis of MASLD, including traditional B-mode ultrasound, Doppler, elastography, and quantitative ultrasound (QUS). QUS analyzes raw, unprocessed radiofrequency (RF) data returned from tissue by inferring the mechanical properties of the tissue through the interaction of the ultrasound beam with the tissue and applying appropriate modeling (*Tamaki, Ajmera & Loomba, 2021*). QUS can model various physical properties, including sound backscattering, sound attenuation, and speed of sound, to produce objective, absolute indicators for quantifying hepatic steatosis.

### Hepatorenal index

In conventional ultrasound, the diagnosis of fatty liver is typically made by comparing the echoes of the liver and the right kidney on the same image. However, this method is affected by various factors including gain, depth, power, and patient anatomy. The hepatorenal index (HRI), a semi-quantitative biomarker of steatosis, improves upon this method by dividing the signal intensity of the liver by that of the renal cortex on the same ultrasound image (*Pirmoazen et al., 2022*). Higher HRI values indicate increased liver echogenicity, corresponding to higher steatosis levels. *Johnson et al. (2021)* demonstrated that an HRI of ≥1.4 is associated with a positive predictive value of over 95% for ≥10% steatosis. However, no corresponding guidelines currently exist. HRI is usually obtained by manually setting the region of interest (ROI) to avoid vascular and focal lesions. However, it cannot be used in the presence of ectopic kidneys or the absence of the right kidney.

### Attenuation coefficient

The attenuation of sound waves in fatty liver tissue differs from that in normal liver parenchyma. The attenuation coefficient (AC) quantifies the rate of energy loss of sound waves as they propagate through tissue, which depends on the wave's frequency and the tissue characteristics (*Ozturk et al., 2023*). Several algorithms are available for estimating AC, including the controlled attenuation parameter (CAP), which is the most widely studied ultrasound technique for quantifying fatty liver. CAP is measured in dB/m, with higher values indicating more severe liver steatosis. CAP is particularly sensitive in detecting fatty livers with more than 10% steatosis (*Pirmoazen et al., 2020*). *Karlas et al. (2017)* demonstrated that the diagnostic performance of CAP was reflected by an AUC value ranging from 0.823 to 0.882. However, CAP measurements are obtained without visualizing the liver, meaning they may be influenced by masses, vessels, or uneven steatosis, which can affect the accuracy of the results. Its accuracy is also diminished in the presence of obesity, ascites, and advanced fibrosis. Therefore, CAP should be combined with other ultrasound techniques to improve diagnostic reliability. In response to these limitations, researchers have developed methods for measuring liver fat content in real

time on conventional ultrasound images, with the ability to accurately localize the region of interest. These methods, such as ultrasound-guided attenuation parameters (UGAP), attenuation coefficients (ATT), and attenuation imaging (ATI), utilize similar principles of attenuation measurement. For example, ATI has demonstrated greater accuracy than CAP, with an AUC ranging from 0.79 to 0.97 (*Zeng et al., 2023*). Despite their promising performance, UGAP, ATT, and ATI techniques are limited by factors such as operator dependence, image quality, and a lack of standardization, which can affect diagnostic accuracy.

### Backscattering coefficient

The backscattering coefficient (BSC) is a quantitative value that reflects the scattering of an ultrasound pulse back to the echo probe (*Wear et al., 2022*). The number of scattered backward-facing ultrasound pulses increases with higher levels of liver fat because fat vesicles in liver cells enhance the scattered ultrasound signal. *Lin et al. (2015)* demonstrates that the backscatter coefficient (BSC) shows strong diagnostic performance in identifying hepatic steatosis, with an AUC of 0.98 for diagnosing steatosis defined by MRI-PDFF $\geq$5%. However, its sensitivity and specificity may vary depending on factors like BMI, and it requires specialized equipment and software for analysis.

### Speed of sound

Speed of sound (SoS) can be used to characterize tissue properties based on changes in ultrasonic echo velocity in various media (*Ferraioli & Monteiro, 2019*). The speed of sound decreases with increasing liver fat content. *Dioguardi Burgio et al. (2018)* demonstrated that the novel ultrasound-based SoS provides high diagnostic performance for detecting and grading hepatic steatosis, with an AUC of 0.882 for detecting any grade of steatosis and 0.989 for moderate to severe steatosis. However, SoS is susceptible to confounding factors such as inflammation, parenchymal edema, and temperature changes.

### Speckle statistics

Speckle statistics, also known as ultrasound envelope statistical parametric imaging (*Park et al., 2022*), analyze the speckle pattern caused by the scattering of ultrasound signals from tissue microstructures, which appears in the ultrasound image. Speckle statistics are based on the parameterization of ultrasound speckle patterns using an established statistical model that describes the scattering properties of the tissue. Acoustic structure quantification (ASQ) was introduced as a more advanced approach that builds upon the principles of speckle statistics. ASQ improves upon traditional methods by directly analyzing the backscattered ultrasound signals, offering more precise and reliable quantification of tissue composition. *Lin et al. (2019)* indicate that, among different scanning planes, the intercostal approach using ASQ achieves the highest AUC value of 0.92. ASQ improves precision over traditional speckle-based methods, but it may be less accessible and more dependent on equipment and scanning conditions compared to CAP.

### Shear wave elastography indicators

Shear wave elastography (SWE) is an imaging technology that generates shear waves in tissues by emitting acoustic radiation pulses and converts the propagation speed of these

waves into corresponding tissue hardness (*Ozturk et al., 2021*). Real-time, two-dimensional shear wave elastography images can be obtained through SWE, and the value of Young's modulus (measured in kPa) of the liver can be quantitatively detected, reflecting the absolute hardness of the liver. The higher the SWE value, the harder the tissue (*Taru et al., 2023*). The modulus of elasticity or stiffness of diseased tissues often differs from that of normal tissues, allowing SWE to distinguish between normal and abnormal tissues. SWE is primarily used to assess liver fibrosis, an important component of MASLD and other liver diseases. It is useful in identifying patients with advanced fibrosis or cirrhosis (*Castera, Friedrich-Rust & Loomba, 2019*). However, for assessing steatosis, SWE showed poor correlation with steatosis grades and did not distinguish between different steatosis grades effectively (*Wang et al., 2025*). On the other hand, SWE could be used alongside other diagnostic tools like CAP to provide a comprehensive assessment of liver health.

### Multimodal ultrasound improves the accuracy of MASLD diagnosis

The development of quantitative ultrasound holds significant promise for liver disease diagnosis, yet the establishment of standardized values remains a critical challenge, influenced by various patient and technical factors. For example, *Shi et al. (2019)* incorporated shear wave attenuation, shear wave absorption, elasticity, dispersion slope, and echo attenuation, achieving an impressive AUC of 0.93. While this result suggests a high degree of accuracy, the study's reliance on a complex model may limit its clinical applicability. Similarly, *Labyed & Milkowski (2020)* developed the ultrasound-derived fat fraction (UDFF) by combining attenuation and backscattering coefficients, achieving an AUC range of 0.83 to 0.94 for liver fat quantification. *Dillman et al. (2022)* supported these findings and demonstrated similar results. While UDFF holds significant promise as a reliable, non-invasive tool for liver fat quantification, challenges related to performance variability and the need for methodological standardization must be addressed.

Moreover, quantitative ultrasound techniques have increasingly been integrated with clinical data to improve diagnostic outcomes. Notably, *Newsome et al. (2020)* introduced the FibroScan-AST (FAST) score, a composite score that combines vibration-controlled transient elastography (VCTE), CAP, and aspartate aminotransferase (AST), which demonstrated the best predictive properties for MASH and advanced fibrosis. In external validation, the FAST score achieved AUCs between 0.74 and 0.95, with sensitivity potentially compromised at lower AUC thresholds. More recently, the American Association for the Study of Liver Diseases (AASLD) (*Rinella et al., 2023b*) recommended a stepwise approach, beginning with the FIB-4 index score, followed by liver stiffness measurement (LSM) *via* VCTE, as the initial method for identifying high-risk MASH. While this recommendation aligns with current evidence, it highlights the challenge of developing a universally applicable screening protocol, as LSM's effectiveness varies with liver condition and technology.

Quantitative ultrasound shows strong potential for diagnosing MASLD, but challenges remain in achieving consistency due to variability in equipment, operator skill, and patient factors. While techniques like UDFF and the FAST score show high accuracy, their performance varies, emphasizing the need for better standardization. Combining

ultrasound with clinical data is promising, but creating universally applicable protocols remains challenging. Further research is needed to enhance these methods for wider clinical use.

## AI in diagnosis of MASLD based on ultrasound (Table 1)

The diagnosis of MASLD using artificial intelligence (AI) is primarily achieved through machine learning (ML). ML can be further divided into supervised learning, unsupervised learning, and deep learning. Supervised learning involves creating predictive models based on input and output data, while unsupervised learning focuses on grouping and pattern recognition using only input data (*De Bruijne, 2016*). Deep learning, a subset of ML, utilizes neural networks to analyze large datasets (*Dinani, Kowdley & Noureddin, 2021*) (Fig. 2). Supervised learning techniques include linear regression, logistic regression, decision trees, K-nearest neighbors, support vector machines, random forests, naive Bayes classification, and gradient boosting, among others. Traditional ML methods can classify and diagnose conditions using input data, which can be either features or raw data. Features are quantifiable data variables derived from expert knowledge that accurately describe the data from the ROI. When raw data is used as input, the algorithm must identify the features autonomously. To compare the performance of different ML methods and the diagnoses made by diagnostic sonographers, several classification metrics are employed. Key indicators include accuracy, sensitivity, specificity, and the area under the curve (AUC) from receiver operating characteristic (ROC) analysis. Below are some major ML methods and related studies.

### Bayes classifier

The Bayes classifier is a probabilistic method widely used in medical diagnostics due to its ability to handle uncertainty and small datasets. In *Ribeiro, Tato Marinho & Sanches (2014)*, a Bayes classifier was used to detect hepatic steatosis from ultrasound images, achieving 93.33% accuracy, with 94.59% sensitivity and 92.11% specificity. Similarly, *Hwang & Cho (2023)* applied a Bayes latent model to identify MASLD predictors, finding that ultrasound attenuation imaging (ATI) was the most effective for predicting hepatic steatosis, with an AUC of 0.923, a sensitivity of 90.2% and specificity of 85.4%. Both studies demonstrate the Bayes classifier's potential in accurately diagnosing MASLD and hepatic steatosis non-invasively. However, the effectiveness of Bayes classifiers may be limited when there is a strong correlation or high dimensionality among features.

### Support vector machine

Support vector machine (SVM) is a type of linear classifier used for supervised learning, where it separates data into different classes by finding the optimal hyperplane. *Basavarajappa et al. (2021)* applied SVM to six ultrasound imaging measurements, including H-mode ultrasound, and found it achieved the highest accuracy with H-mode data. Similarly, *Nagy et al. (2015)* used SVM to classify hepatic steatosis in 228 patients, showing that the coefficient of variation of luminance was most effective for distinguishing mild and moderate-to-severe steatosis. In summary, SVM excels in high-dimensional classification tasks but requires careful feature selection and parameter tuning to avoid

**Table 1  ML algorithm for diagnosis of MASLD patients based on ultrasound images.**

| Author | Years | Sample size | Classification categories | Parameters in the model | ML algorithm | Results | Reference standard | Data preprocessing /augmentation |
|---|---|---|---|---|---|---|---|---|
| *Ribeiro, Tato Marinho & Sanches (2014)* | 2014 | 74 patients | Normal, steatosis | Textural features | CAD (Bayes) | Acc = 0.933 Sen = 0946 Spec = 0.921 | / | Radiofrequency image estimation; image decomposition; speckle and despeckle separation; normalization and standardization; feature set comparison |
| *Hwang & Cho (2023)* | 2023 | 89 children | Normal, steatosis | ATI | Bayes classifier | Sen = 0.894 Spec = 1 | / | Envelope estimation; speckle decomposition |
| *Basavarajappa et al. (2021)* | 2021 | 21 rats | Normal liver and mild, severe NAFLD | Multiparametric ultrasound | SVM | Acc = 0.92 | Pathology | Z-score normalization; principal component analysis |
| *Nagy et al. (2015)* | 2015 | 228 patients | None, mild, moderate, and severe steatosis | CVL | SVM | AUC = 0.923 Sen = 0.813 Spec = 0.89 | Biopsy | Intensity histogram analysis |
| *Tang et al. (2018)* | 2018 | 60 rats | None, mild, moderate, and severe steatosis | Elastography +QUS | RF | AUC = 0.66 (mild) AUC = 0.84 (moderate) AUC = 0.87 (severe) | Pathology | Echo envelope extraction |
| *Destrempes et al. (2022)* | 2022 | 82 patients | None, mild, moderate, and severe steatosis | Elastography +QUS | RF | AUC = 0.90(mild) AUC = 0.81(moderate) AUC = 0.78 (severe) | Biopsy | Echo envelope extraction; compensation for time gain compensation; sliding window technique; Winsorization |
| *Mihai Mihailescu (2013)* | 2013 | 120 patients | Stage of steatosis | Minimum and maximum attenuation, median gray levels | RF | Acc = 0.908 | / | Robust brightness estimation |
| *Acharya et al. (2012)* | 2012 | 100 images | Normal, fatty liver | Textural features | CAD (DT) | Acc = 0.933 | / | Image standardization |
| *Subramanya et al. (2014)* | 2014 | 53 images | Normal, fatty liver | Texture features | CAD (SVM) | Acc = 0.849 | / | Feature combination; feature selection |
| *Saba et al. (2016)* | 2016 | 62 patients | Normal, fatty liver | Texture features | CAD (BPN) | Acc = 0.976 Sen = 0981 Spec = 0.972 | / | Standardization; feature combination and Scaling |
| *Acharya et al. (2016)* | 2016 | 150 images | Normal, fatty liver and cirrhosis | Texture features | CAD (PNN) | Acc = 0.973 Sen = 1 Spec = 0.960 | / | Morphological processing; image resizing and contrast enhancement; curvelet transform; feature reduction |
| *Kuppili et al. (2017)* | 2017 | 63 patients | Normal, fatty liver | Texture features | CAD (ELM) | Acc = 0.968 AUC = 0.97 | Biopsy | Standardization; data subsampling; |
| *Biswas et al. (2018)* | 2018 | 63 patients | Normal, fatty liver | Features | CNN | Acc = 0.92 | Biopsy | Image optimization; image cropping and border stripping |

**Notes.**

Sen, Sensitivity; Spec, Specificity; Acc, Accuracy; AUC, Area Under Curve.

overfitting or underfitting, and can be computationally intensive with large datasets. Despite these challenges, SVM remains a powerful tool in the diagnosis of MASLD due to its robustness and ability to handle complex data.

### Random Forest

Random Forest (RF) is a classifier that uses multiple decision trees to improve accuracy and reduce overfitting by averaging the results from several trees. *Tang et al. (2018)* used RF to classify liver conditions in rats, finding that combining QUS and shear-wave elastography improved accuracy over elastography alone, though human data are still needed to confirm these results. *Destrempes et al. (2022)* demonstrated this view on human data. *Mihai Mihailescu (2013)* compared RF and SVM in assessing MASLD severity,

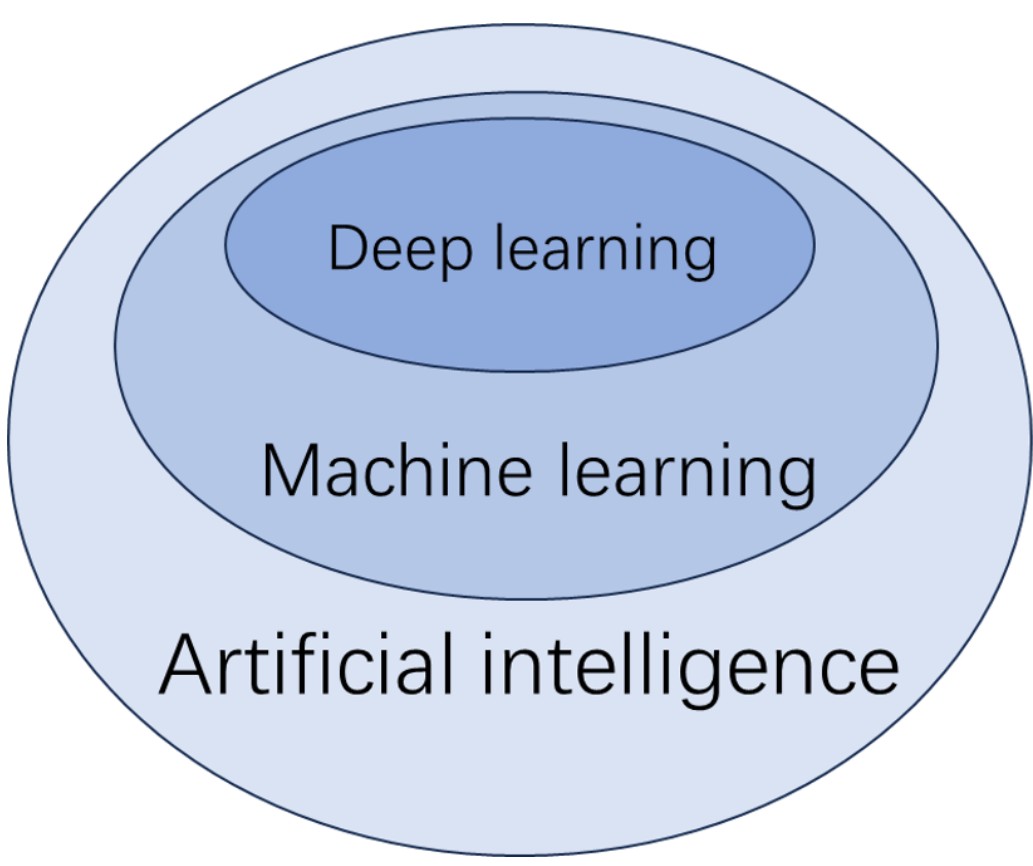

**Figure 2 Deep learning.**

demonstrating that RF performed better in terms of accuracy without the need for feature selection. In conclusion, RF is effective in improving classification accuracy, especially when combining multiple features. Its main advantages are high accuracy and resistance to overfitting. However, it can be computationally expensive and may require careful tuning of hyperparameters for optimal results.

### Deep learning

Deep learning (DL) is a type of machine learning (ML) that builds on the development of artificial neural networks. It outperforms traditional ML, brings ML closer to Artificial Intelligence, and holds the promise of being far superior to previous techniques (Table 2). DL is not affected by image variations, does not ROIs, is not limited to expert-defined features, and can be trained on a large number of images. The algorithm uses multilayer neural networks for detection, classification, and segmentation of biomedical images. Convolutional neural networks (CNNs) are a special type of neural network in deep learning, primarily consisting of convolutional layers, pooling layers, and fully connected layers (*Hosny et al., 2018*) (Fig. 3). They are mainly used to process data with a grid

**Table 2  DL algorithm for diagnosis of MASLD patients based on ultrasound images.**

| Author | Years | Sample size | Classification categories | DL algorithm | Results | Reference standard | Data preprocessing/ augmentation | Pros and cons |
|---|---|---|---|---|---|---|---|---|
| *Cao et al. (2019)* | 2019 | 240 participants | Normal liver and mild, moderate, and severe NAFLD | CNN | AUC = 0.958 | / | Image resizing; envelope signal quantification; data expansion | High diagnostic accuracy, especially for severe NAFLD cases. Struggles with mild/moderate distinction. |
| *Yang et al. (2023)* | 2023 | 928 participants (1,856 images) | None, mild, moderate, and severe steatosis | 2S-NNet | AUC≥0.90 (mild) AUC ≥ 0.85 (moderate) AUC≥0.85 (severe) | / | Image stitching | A novel two-section deep learning model. Offers a solution for large-scale population screening. |
| *Cha et al. (2021)* | 2021 | 294 participants | Normal and mild fatty liver | DCNN | ICC = 0.734 | / | Image resizing | A novel automated HRI quantification method. Unvalidated generalizability to severe disease. |
| *Zsombor et al. (2023)* | 2023 | 102 patients | None, mild, moderate, and severe steatosis | CNN | AUC = 0.758 (mild) AUC = 0.803 (moderate/severe) | MRI-PDFF | Data normalization; various image transformations such as rotations and flips | Easy to implement in clinical practice. Lacks histology validation for comparison. |
| *Nguyen et al. (2021)* | 2021 | 60 rabbits | Normal, steatosis | CNN | Acc = 0. 738 | Pathology | Removal of noisy data and outliers; resizing; normalization; regularization techniques | Simplifying clinical workflow. Accuracy affected by liver fibrosis. Small dataset leading to variability in test results. |
| *Han et al. (2020)* | 2020 | 204 participants | NAFLD, no NAFLD | 1D-CNN | Acc = 0. 96 AUC = 0.958 | MRI-PDFF | remove noise and artifacts; normalization; random transformations (*e.g.*, scaling, rotation) | Robust to system setting changes. Potential saturation effect for high fat fractions. Limited generalizability due to single platform and operator |
| *Sanabria et al. (2022)* | 2022 | 31 patients | None, mild, moderate, and severe steatosis | 2D-CNN, 3D-CNN | AUC≥0.90 | MRI-PDFF | Logarithmic compression; denoising; normalization; extracting multiple patches | Uses raw data for enhanced diagnostics. Limited by available ultrasound machines and small patient sample. |
| *Jeon et al. (2023)* | 2023 | 173 participants | Normal liver and mild, moderate, and severe NAFLD | 2D-CNN | AUC = 0.97 | MRI-PDFF | Envelope extraction; logarithmic compression; noise reduction; normalization; extracting multiple patches | QUS parametric maps and B-mode images for diagnosing. Limited by single-center data, saturation in severe steatosis. |
| *Vianna et al. (2023)* | 2023 | 199 patients | None, mild, moderate, and severe steatosis | VGG16 | AUC = 0.98 (mild) AUC = 0.67 (moderate) AUC = 0.66 (severe) | Biopsy | Image cropping; image resizing; standardization | Outperformed most radiologists in detecting steatosis. Limited generalizability due to single-center study and no data augmentation. |
| *Liu et al. (2024)* | 2024 | 710 participants | None, mild, moderate to severe steatosis | VGG16 | AUC = 0.85 (mild) AUC = 0.95 (moderate to severe) | / | Image resizing; normalization; random transformations (*e.g.*, scaling, rotation) | The new multi-input model showed significant improvement. |
| *Che et al. (2021)* | 2021 | 55 patients | Normal, fatty liver | 2D-ResNet | AUC = 0.978 | Biopsy | Image cropping; image resizing; random transformations (*e.g.*, scaling, rotation) | Utilizes advanced feature fusion techniques and multi-scale analysis. Limited by dataset size. Dependence on quality data. |
| *Chou et al. (2021)* | 2021 | 2070 patients (21855 images) | None, mild, moderate, and severe steatosis | ResNet-50 v2 | AUC = 0.974 (mild) AUC = 0.971 (moderate) AUC = 0.981 (severe) | / | Image cropping; image resizing; normalization; random transformations (*e.g.*, scaling, rotation) | A large dataset for better accuracy. Limited by variations in image quality, motion artifacts, and regional bias from a single hospital dataset. |

| Author | Years | Sample size | Classification categories | DL algorithm | Results | Reference standard | Data preprocessing/ augmentation | Pros and cons |
|---|---|---|---|---|---|---|---|---|
| *Zamanian et al. (2021)* | 2021 | 55 patients | Normal, fatty liver | ResNet101+ SVM | Acc 0.986 AUC = 0.9998 | Biopsy | Image resizing; color modifications; random transformations (*e.g.*, scaling, rotation) | Enhances model generalization and reduces overfitting for improved accuracy. Relies heavily on data augmentation |
| *Hardy et al. (2023)* | 2023 | 55 patients | Normal, fatty liver | ResNet-50 | AUC = 0.904 | Biopsy | Center cropping; linear scaling; random resized crop | Enhances classification accuracy with synthetic data. Limited by small dataset size. |
| *Constantinescu et al. (2020)* | 2020 | 60 patients | Normal, steato­sis | Inception-v3 | Acc = 0.932 AUC = 0.93 | / | Image cropping; image rescaling; random trans­formations (*e.g.*, scal­ing, rotation); dropout, activity regularization, kernel regularization | Reduces the need for large datasets by using fine-tuning. Limited by small dataset size and no comparison with expert diag­noses. |
| *Santhosh Reddy, Bharath & Rajalakshmi (2018)* | 2018 | 157 images | Normal, fatty liver | VGG-16 | Acc = 0.96 | / | Image cropping; image resizing; random trans­formations (*e.g.*, scaling, rotation) | Improves diagnosis efficiency with limited data. Limited by small dataset size and no com­parison with expert diagnoses. |
| *Byra et al. (2021)* | 2021 | 135 participants | Normal, fatty liver | ResNet-50 | AUC 0.91 | MRI-PDFF | Image cropping; image resizing; image shiftin; horizontal flipping | Achieves high diagnostic perfor­mance with multiple liver views. Limited by a small dataset and potential bias. |
| *Li et al. (2022)* | 2022 | 3,310 patients | None, mild, moderate, and severe steatosis | ResNet18 | AUC = 0.85 (mild) AUC = 0.91 (moderate) AUC = 0.93 (severe) | Biopsy | Automatic cropping; viewpoint filtering; multi-scanner inclusion; multi-view Inclusion | Generalizable across multiple scanners and viewpoint. The model may be sensitive to the imaging protocol and dataset biases. |

**Notes.**
AUC, Area Under Curve; Acc, accuracy; ICC, Intraclass Correlation Coefficient.

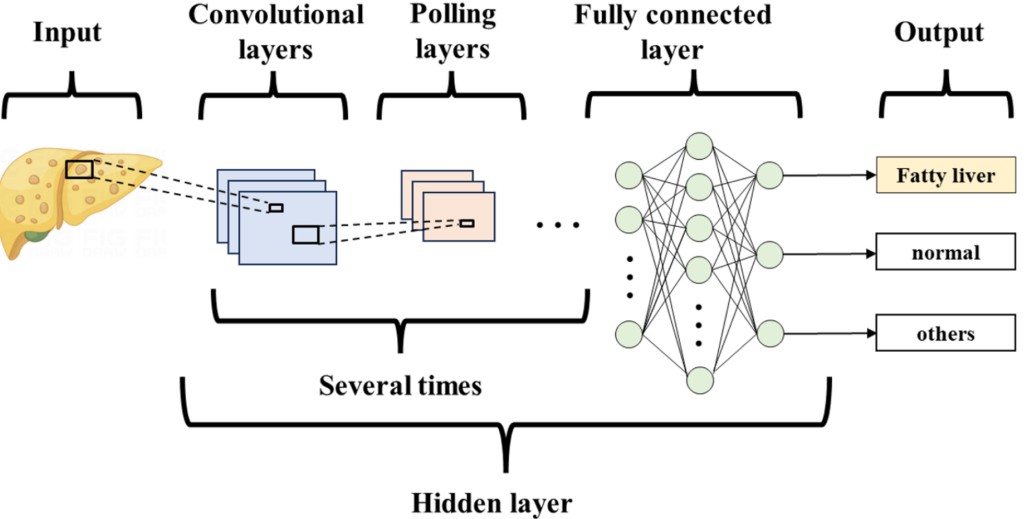

**Figure 3  Convolutional neural network.**

structure, extract feature maps, and perform feature aggregation. Currently, CNNs are the most popular type of architecture in medical imaging.

*Deep learning based on feature acquisition.*  DL has shown promise in extracting features from ultrasound images to aid in diagnosing MASLD. *Cao et al. (2019)* analyzed liver

images from 240 patients using envelope signals, gray signals, and a DL index derived from three image processing technologies. The DL index outperformed the envelope and gray values in diagnostic performance. Similarly, *Yang et al. (2023)* developed and validated a DL system using ultrasound images from 928 subjects. Their system, which employed a two-segment neural network (2S-NNet), classified the severity of hepatic steatosis based on three diagnostic features: bright liver, blurred intrahepatic catheter, and impaired visibility of more than half of the diaphragm. Both studies found high accuracy for diagnosing MASLD, though accuracy was lower for mild and moderate cases. However, the studies have limitations: the limited sample size of *Cao et al. (2019)* further restricts the broader applicability of their findings. *Yang et al. (2023)* lacked secondary validation (*e.g.*, MRI or biopsy), and sonographer training was not standardized, which could affect the results. Therefore, while DL shows potential for diagnosing MASLD, further validation with standardized protocols is necessary to confirm its reliability.

*Deep learning based on hepatorenal index.*  Many DL studies of MASLD have utilized the HRI. *Cha et al. (2021)* analyzed 294 ultrasound images from living liver donors and found that an HRI automatically quantified using a deep convolutional neural network (DCNN) showed high consistency with measurements made by ultrasound diagnosticians. However, this study focused only on normal and mild fatty liver, excluding moderate and severe cases, which limits its broader applicability. In contrast, *Zsombor et al. (2023)*, in a single-center prospective study, diagnosed mild and moderate steatohepatitis in 102 subjects using an AI-based liver and kidney index (AI-HRI) calculated by DCNN. Their results showed a higher AUC, higher sensitivity, and lower specificity compared to manually measured indices from previous studies. AI-HRI also outperformed grayscale ultrasound in detecting mild steatosis. Nevertheless, AI-HRI should not replace grayscale ultrasound, as a combination of both methods is necessary to enhance diagnostic accuracy. Furthermore, the sample size in this study was limited, and further validation in a multicenter setting is needed to confirm the findings.

*Deep learning based on raw data.*  The current study indicates that diagnosing MASLD using raw data may offer advantages over traditional quantitative ultrasound (QUS) indicators. *Nguyen et al. (2021)* developed a rabbit MASLD model and showed that a one-dimensional CNN processing raw radiofrequency signals achieved 74% accuracy, outperforming the 59% accuracy of a QUS-SVM model. However, as this study was animal-based, its findings require further validation in clinical settings. *Han et al. (2020)* reported a 96% classification accuracy using a one-dimensional CNN with raw ultrasound radiofrequency signals from 102 subjects. Additionally, studies by *Sanabria et al. (2022)* and *Jeon et al. (2023)* highlighted that two-dimensional CNNs outperform one-dimensional models for classifying hepatic steatosis using RF data, showing superior diagnostic performance over traditional methods. The combination of raw radiofrequency data and deep learning holds significant promise for improving MASLD diagnosis by providing more accurate liver fat quantification. However, its clinical applicability is still limited by

challenges such as data standardization, validation, and the need for large-scale patient datasets.

*Transfer learning in deep learning.* Deep learning has achieved great success in image recognition due to its ability to learn hierarchical features from large datasets. However, training deep models from scratch requires vast amounts of labeled data and significant computational resources. Transfer learning addresses this issue by leveraging pre-trained models, that have been trained on large, well-established datasets like ImageNet (*Morid, Borjali & Del Fiol, 2021*). These pre-trained models learn general features (*e.g.*, edge detection, texture recognition) that can be reused for new, related tasks (*Cheplygina, de Bruijne & Pluim, 2019*). As a result, transfer learning allows deep learning models to perform effectively even on smaller, domain-specific datasets.

Pre-trained models like VGG-16 (*Vianna et al., 2023*; *Liu et al., 2024*), ResNet (*Che et al., 2021*; *Chou et al., 2021*; *Zamanian et al., 2021*; *Hardy et al., 2023*) and Inception (*Constantinescu et al., 2020*) have been applied to liver condition classification, improving diagnostic accuracy with less labeled data. For instance, *Byra et al. (2018)* demonstrated that a ResNet-v2 model, fine-tuned for liver ultrasound images, outperformed traditional methods, showing higher AUC scores in identifying fatty liver. Similarly, *Santhosh Reddy, Bharath & Rajalakshmi (2018)* utilized a VGG-16 pre-trained model and achieved 90.6% accuracy in classifying fatty liver from ultrasound images, showcasing the effectiveness of transfer learning in improving diagnostic performance in smaller datasets. *Byra et al. (2021)* and *Li et al. (2022)* both demonstrated that multi-view ultrasound enhances diagnostic accuracy, with *Byra et al. (2021)* finding the right posterior portal vein to be the most accurate, while *Li et al. (2022)* showed that combining multi-view data from 3,310 patients using ResNet-18 achieved performance comparable to CAP measurements and high consistency across ultrasound scanners.

In recent years, different teams have optimized datasets (*e.g.*, expanding dataset size, acquiring ultrasound images with different instruments) and compared various pre-trained models to identify the most suitable model, but no unified conclusion has been reached. Despite its success, transfer learning is not without limitations. One challenge is the domain gap, where the source data (such as general images used to train pre-trained models) may differ significantly from the target data (*e.g.*, medical images). This discrepancy can result in suboptimal performance if not properly addressed, as the model might struggle to generalize from one domain to another (*Hosseinzadeh Taher et al., 2025*). Additionally, fine-tuning is needed to avoid overfitting with smaller datasets (*Xie et al., 2021*). The effectiveness of pre-trained models depends on their alignment with the specific task, requiring adjustments for optimal performance in medical imaging (*Atasever et al., 2023*).

Despite these challenges, the increasing availability of pre-trained models and the advancement of transfer learning techniques continue to enhance diagnostic accuracy, solidifying transfer learning as a leading approach in medical image analysis. In conclusion, transfer learning is a transformative technique within deep learning. By reusing knowledge from pre-trained models, transfer learning significantly improves performance while reducing the need for large-scale training datasets. As the field evolves, integrating transfer

learning with emerging technologies, such as synthetic data generation and multi-modal learning, promises to further enhance the accuracy and robustness of diagnostic systems.

### Automated machine learning

Recently, a new method in machine learning, known as automated machine learning (AutoML), has been proposed. AutoML automates key aspects of the machine learning process, including data preprocessing, optimal algorithm selection, and hyperparameter tuning, significantly reducing the time required to build machine learning models (*Wever et al., 2021*). This automation allows for faster model development, making it more accessible to non-experts and streamlining the workflow for experienced practitioners. *Tahmasebi et al. (2023)* conducted a study involving 120 subjects, both with and without MASLD, using MRI-PDFF as the reference criterion. They employed AutoML Vision to develop a supervised machine learning model for assessing MASLD, achieving a sensitivity of 72.2% and a specificity of 94.6%. This study highlights AutoML's potential in medical diagnostics, demonstrating its ability to build accurate models for complex tasks like assessing MASLD. However, despite these promising results, there is still a scarcity of relevant studies utilizing AutoML in this field, which suggests a need for further research to fully explore its capabilities and limitations.

### Computer-aided design based on feature input

Computer-aided design (CAD) techniques have become increasingly integral in enhancing the diagnosis of non-alcoholic fatty liver disease (MASLD) through ultrasound imaging. The incorporation of ML in CAD systems has shown significant promise in analyzing liver ultrasound images with greater accuracy and efficiency than traditional methods. As a result, researchers have focused on developing the most effective CAD systems, exploring various classifiers to identify the best-performing models for diagnosing MASLD.

Early studies, such as those by *Acharya et al. (2012)*, employed the Symtosis CAD method to extract features from liver ultrasound images, combining texture analysis, higher-order spectra, and wavelet transforms. This technique, paired with a decision tree (DT) classifier, accurately distinguished between normal and fatty liver images. Similarly, *Subramanya et al. (2014)* used a SVM-based CAD system to diagnose fatty liver, finding that Laws ratio features yielded the best classification results.

As technology advanced, CAD systems evolved, incorporating more sophisticated algorithms and a broader range of features. *Saba et al. (2016)* applied Symtosis to extract additional features, including Haralick, Gabor, and Fourier transforms, improving the system's ability to capture subtle liver texture changes. Their study showed that a back propagation neural network (BPNN) classifier outperformed traditional DT classifiers. *Acharya et al. (2016)* introduced a curvelet-wave transform method with a probabilistic neural network (PNN), which showed high classification accuracy using only six features. These advancements marked a shift toward more efficient and robust models, reducing the complexity of the feature set required for accurate classification.

Further improvements were made in more recent studies. *Kuppili et al. (2017)* and *Biswas et al. (2018)* explored the combination of Symtosis with extreme learning machines (ELM) and CNNs, both of which led to faster and more accurate liver image classification.

These models capitalized on the ability of CNNs to automatically learn and refine features from raw data, removing the need for extensive manual feature extraction.

While CAD systems using various classifiers have been developed for diagnosing hepatic steatosis, several challenges remain. Factors such as image quality, feature selection, and the complexity of classification algorithms can all impact the performance of these systems. The application of machine learning techniques like SVM, CNN, and PNN has significantly improved liver image classification accuracy. The future of ultrasonic CAD holds great promise, particularly in enhancing the early detection and management of MASLD, offering more reliable and efficient diagnostic tools.

## SUMMARY AND PROSPECTIVES

The application of AI in MASLD has gained significant attention. Currently, ultrasound is the most widely used tool for MASLD screening, and AI-assisted ultrasound has shown considerable promise. A meta-analysis (*Decharatanachart et al., 2021*) demonstrated that AI-assisted ultrasound achieved a sensitivity of 0.97, specificity of 0.98, positive predictive value (PPV) of 0.98, negative predictive value (NPV) of 0.95, and area under the curve (AUC) of 0.98, outperforming AI-assisted clinical datasets. Additionally, neural networks have shown superior performance compared to non-neural network models. However, most studies to date rely on retrospective, single-center data, which introduces selection bias, limits generalizability, and results in variability in diagnostic tools and operational procedures.

AI offers key advantages over traditional statistical models, such as identifying complex patterns, integrating multiple factors, and creating predictive models. These models aid risk stratification, improve diagnostic accuracy, and enhance patient outcomes (*Dinani, Kowdley & Noureddin, 2021*). In MASLD diagnosis, AI utilizes diverse data sources, including electronic health records, laboratory tests, imaging, and liver histopathology data (*Li et al., 2021*). AI-assisted ultrasound improves diagnostic accuracy, reduces reliance on operator experience, and minimizes subjectivity. Additionally, AI can enhance diagnostic efficiency, lower costs, and reduce the burden on sonographers without replacing them. Its integration is expected to significantly impact primary care, telemedicine, clinical decision support systems, and early intervention for disease progression.

Despite its promising benefits, current research on AI in MASLD diagnosis faces several limitations. First, many studies rely on retrospective, single-center data, typically characterized by small sample sizes and limited representativeness, leading to potential selection bias and reduced generalizability. Variability in diagnostic tools and operational procedures across settings also contributes to inconsistencies in model performance. Furthermore, AI models are often overfitted to training data, performing well on the data they were trained on but poorly when applied to new, unseen data. The absence of external validation exacerbates this issue. To address these challenges, AI models must undergo robust validation in multi-center, prospective studies involving diverse patient populations to ensure their consistent performance across various settings.

Another significant challenge is the lack of interpretability, commonly referred to as the "black box" problem. Many advanced AI models, particularly deep learning

systems, process data through complex algorithms that are difficult for human experts to interpret. While these models often yield accurate predictions, they do so without providing clear explanations of how decisions are made. This lack of transparency is particularly problematic in healthcare, where clinicians must understand and justify diagnostic decisions. As a result, clinicians may hesitate to rely on AI recommendations, especially when they cannot validate the underlying logic behind a diagnosis or treatment suggestion. Furthermore, this lack of interpretability raises significant legal and ethical concerns. If an AI model makes a diagnostic error that harms a patient, determining accountability—whether it lies with the developer, the clinician using the tool, or the institution implementing it—becomes difficult. Enhancing explainable AI, which aims to make AI models more transparent and understandable, is crucial for addressing these concerns and promoting the adoption of AI in clinical settings.

The ethical implications of AI in healthcare are multifaceted, raising concerns about data privacy, bias, and accountability. AI systems require large volumes of sensitive patient data to function effectively, raising significant concerns about how this data is collected, stored, and utilized. Ensuring compliance with data protection regulations is critical for safeguarding patient privacy. As AI models become increasingly integrated into clinical decision-making, secure handling of data is essential to maintain patient trust and ensure system integrity. One of the most pressing ethical concerns is the potential for AI models to inherit biases from the data on which they are trained. If training datasets are not diverse or representative of global populations, AI systems could perpetuate or even exacerbate existing healthcare disparities. To mitigate these biases, it is essential that AI training datasets reflect a broad range of ethnicities, ages, genders, and socioeconomic backgrounds. The integration of AI in clinical practice also raises complex questions about responsibility and accountability. In the event of diagnostic errors or adverse patient outcomes caused by AI-assisted tools, it may be unclear who should be held accountable. This lack of clarity could undermine trust in AI systems and complicate medical malpractice frameworks. These concerns underscore the need for clear ethical guidelines and frameworks to govern AI deployment in healthcare.

The future potential of AI in MASLD diagnosis is vast, particularly with the integration of multi-modal data. In addition to traditional ultrasound images, AI is expected to incorporate clinical data, laboratory test results, and multi-omics information to further enhance diagnostic accuracy. Future research should focus on multi-center, prospective studies to gather high-quality, representative data and address current limitations in data quality. Moreover, there is a need to develop more transparent and interpretable models to improve clinicians' trust in AI. Beyond early screening for MASLD, AI-assisted ultrasound may expand to diagnose other liver diseases, monitor disease progression, and evaluate treatment efficacy. With continued advancements, AI has the potential to play a crucial role in primary care and telemedicine, providing accessible, cost-effective diagnostic tools that support early intervention and precision treatment for liver diseases globally.

### Funding
This work was funded by the Science and Technology Bureau of Quanzhou (grant number 2024QZC009YR) and the Fujian Provincial Health Technology Project (grant number 2022CXB010). The funders had no role in study design, data collection and analysis, decision to publish, or preparation of the manuscript.

### Grant Disclosures
The following grant information was disclosed by the authors:
Science and Technology Bureau of Quanzhou: 2024QZC009YR.
The Fujian Provincial Health Technology Project: 2022CXB010.

### Competing Interests
The authors declare there are no competing interests.

### Author Contributions
- Jiawen Li conceived and designed the experiments, analyzed the data, prepared figures and/or tables, authored or reviewed drafts of the article, and approved the final draft.
- Jianhui Chen conceived and designed the experiments, analyzed the data, authored or reviewed drafts of the article, and approved the final draft.
- Xiaohong Zeng performed the experiments, prepared figures and/or tables, and approved the final draft.
- Guorong Lyu performed the experiments, prepared figures and/or tables, and approved the final draft.
- Shu Lin analyzed the data, authored or reviewed drafts of the article, and approved the final draft.
- Shaozheng He conceived and designed the experiments, authored or reviewed drafts of the article, and approved the final draft.

### Data Availability
As this is a review article, it does not involve the generation of new experimental data or original code.

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
