# Peer review of "Update of machine learning for ultrasound diagnosis of metabolic dysfunction-associated steatotic liver disease: a bright future for deep learning"

_PeerJ, doi:10.7717/peerj.19645_

## Round 0.1 · original submission · Major Revisions

Please address the reviewer comments in your revised manuscript.

Reviewer 1 ·

Basic reporting

• Clarity and language: The manuscript is well-written in professional, clear English. However, some sentences are long and would have been much better if they were shortened and expressed directly. Some rewriting may be made regarding conciseness and clarity, such as very wordy descriptions of methodologies.
• Introduction and background: The introduction sets a good basis regarding the diagnosis of NAFLD and the application of AI. However, some references need a bit more contextualization to develop better flow and coherence. Also, the word "NAFLD" is an old term; the new term is “MASLD.”
• Literature citations: The paper is well-referenced, but a few of the key references require expansion with a critical discussion of its importance.
• Structure: The manuscript does meet the standards for PeerJ, except there are redundant details presented that may be combined between certain sections like in the Literature Review - machine learning techniques
• Figures and Tables: Relevant Figures and Tables have appropriate legends appropriately completed

Experimental design

• Scope and relevance: Submission falls within the aims and scope of the Journal. The topic is timely and of relevance to the scientific community.
• Methodology: The methodology of literature selection is clearly presented in the inclusion and exclusion criteria. However, the review does not discuss the potential biases of the selected literature.
• Coverage of the field: The manuscript covers major advances in AI and ML for NAFLD diagnosis. A deeper discussion of the limitations of the existing studies and unresolved challenges would enhance the overall contribution of this study.
• Organization: The manuscript is well-organized, but subsections within deep learning methodologies could be better logically laid out to improve readability.

Validity of the findings

• Logical argumentation: The review develops and supports an argument quite well. However, some sections summarize studies without a critical synthesis of findings.
• Future directions: Although the conclusion does outline the future directions, it would be even better if it were deeper with a discussion of the possible pitfalls in the adoption of AI for clinical applications in ultrasound.
• Limitations: While the paper mentions some limitations with respect to AI diagnosis of NAFLD, these need to be further emphasized.

Additional comments

General Comments for Authors
• Terminology: Change all "NAFLD" to the new term "MASLD"
• Avoid redundancy: Some discussions, like the architecture of CNN, could have been summarized to avoid repetitions of some deep learning methodologies.
• Need for more critical analysis: The manuscript could contain a more critical discussion on the challenges of AI in clinical translation, specifically generalizability, interpretability, and ethical issues.
• Logical flow: Some sections, particularly in methodology and discussion, need re-organization to logical flow.

Reviewer 2 ·

Basic reporting

The manuscript titled “Update of Machine learning for ultrasound diagnosis of nonalcoholic fatty liver disease: a bright future for deep learning” is a review, which intends to comprehensively analyze and comment on the new progress of ultrasound in the diagnosis of non-alcoholic fatty liver disease (NAFLD), especially the development of artificial intelligence in this field. However, the paper only contains a simple list of some literatures in related fields, without in-depth understanding and comparison of different literatures and the author’s objective evaluation about AI research in ultrasonic diagnosis of NAFLD cannot be found.

Experimental design

no comment

Validity of the findings

no comment

Additional comments

There are some issues need to be pointed out:
1. SECTION Survey Methodology (Line64-92) is superfluous. The author does not conduct a meta-analysis and there is no need for a thorough literature search and no need for deduplication.
2. Line115-149, not only has little to do with the theme of the review, but more like a popular science introduction, do you think it can be deleted?
3. Line150 "Ultrasound detection of non-alcoholic fatty liver disease" Is your intention to introduce the quantitative indicators that can be used in the diagnosis of NAFLD by ultrasound? Would you write the title more clearly?
4. Line159-169 Don't you think it is necessary to give some examples of specific values and diagnostic criteria, and comprehensively discuss the sensitivity, specificity and accuracy of HRI in diagnosing NAFLD? The same problem exists in Parts 2.2-2.7 of the manuscript
5. Line171-174 What does this paragraph mean? Do you really think these sentences should appear here?
6. Line199-192 “They use similar principles (attenuation measurements) to measure liver fat and have lower failure rates than CAP and a higher inter-observer reproducibility" Obviously, authors should give specific values for failure rates and inter-observer reproducibility to make content more readable.
7. Line282-285 “Utilizing a two-level Bayes latent model for their analysis, Hwang et al. (Hwang & Cho 2023) found that among non-invasive predictors of NAFLD, ultrasound attenuation imaging was the most effective in predicting hepatic steatosis.” What predictors? Can you list them and show the result of comparative data?

Reviewer 3 ·

Basic reporting

All comments have been added in detail to the last section.

Experimental design

All comments have been added in detail to the last section.

Validity of the findings

All comments have been added in detail to the last section.

Additional comments

Review Report for PeerJ
(Update of Machine learning for ultrasound diagnosis of nonalcoholic fatty liver disease: a bright future for deep learning)

1. In the study, studies on liver disease detection in the literature related to artificial intelligence were examined in depth.

2. In the introduction, Non-alcoholic fatty liver disease, survey methodology, was mentioned at a certain level. At the end of this section, how this review differs from similar reviews in the literature and its basic contributions to the literature should be mentioned in detail.

3. While the Inclusion and Exclusion Criteria and search terms mentioned regarding search methodology are sufficient, it is recommended that the search database should be detailed in future studies, especially in terms of open access publishing.

4. For shared machine learning approach studies related to Non-alcoholic fatty liver disease diagnosis specified in table-1, data preprocessing/augmentation sections should also be added to this table as an additional column.

5. It is suggested that sections including originality, pros and cons, data augmentation and data processing should be added to this table for Non-alcoholic fatty liver disease diagnosis studies shared in the literature using the deep example given in Table-2.

As a result, the study can make an important contribution to the literature, but attention should be paid to the sections listed above.

---

## Round 0.2 · accepted · Accept

Thank you for your efforts in addressing the reviewer feedback. You have now satisfied the requested amendments and your work has been deemed suitable for publication.

Reviewer 3 ·

Basic reporting

All comments have been added in detail to the last section.

Experimental design

All comments have been added in detail to the last section.

Validity of the findings

All comments have been added in detail to the last section.

Additional comments

Review Report for PeerJ
(Update of machine learning for ultrasound diagnosis of metabolic dysfunction-associated steatotic liver disease: a bright future for deep learning)

Both the new changes in this version of the paper and the responses to reviewer comments are generally at an adequate level. Best regards.